# A Constant Round Write-Only ORAM

**Bo Zhao [1,*], Zhihong Chen [1], Hai Lin [1] and XiangMin Ji [2]**

[1]  School of Cyber Science and Engineering, Wuhan University, Wuhan 430072, China;
  chenzhihong@whu.edu.cn (Z.C.); lin.hai@whu.edu.cn (H.L.)

[2]  College of Computer Information Science, Fujian Agriculture and Forestry University, Fuzhou 350002, China;
  jixm@126.com

*  Correspondence: zhaobo@whu.edu.cn

**Abstract:** The write-only oblivious RAM (ORAM) is proposed to efficiently protect the privacy of applications such as cloud storage synchronization and encrypted hidden volumes. For $N$ blocks with size $B = \Omega(log^2N)$, the most efficient write-only ORAM, DetWoORAM, achieves $O(B)$ communication complexity with $O(logN)$ rounds per logical write. We propose a two-level write-only ORAM and achieve $O(B)$ communication complexity with $O(1)$ rounds. Similar to the traditional bucket-based ORAM schemes, we set a rate for the write operation to further reduce the communication complexity. The top-level stores data blocks in a flat array and the write pattern is protected by writing blocks uniformly at random. The second level employs a binary tree to store the position map of data blocks. To avoid recursive storage, a static position map for blocks in the second level is used. Both the analysis and experiments show that, besides the achieved low communication complexity and rounds, the stash sizes in the top level and the second level are bounded to $O(B)$ and $\omega(B)$, respectively.

**Keywords:** write-only ORAM; privacy; access pattern

---

## 1. Introduction

For applications like outsourced storage, a client's data is stored on an untrusted server [1,2]. Even if the data is fully encrypted, sensitive information might still be leaked to the server or adversaries that can observe the entire communication between the client and the server [3]. Oblivious RAM (ORAM) is a protection guaranteeing that the server and such adversaries cannot distinguish an access pattern from another. ORAM has been widely studied and its performance has been improved both in theory [4–8] and in practice [9,10]. In addition to the outsourced storage, it has also been applied in other various applications, such as secure processor [11–13] and secure multi-party computation [14–18].

### 1.1. The Emergence of Write-Only ORAM Schemes and Their Efficiencies

The traditional ORAM schemes protect the privacy of both the read access pattern and write access pattern. However, for some applications such as the encrypted hidden volume [19] and cloud storage synchronization [20], the server is able to see the entire history of write operations, but it cannot see which blocks are being read. In this case, only the write access pattern is required to be protected, which brings a weaker security notion of ORAM. If using the traditional ORAM schemes to these applications, a substantial amount of shuffling is required for a write operation, which incurs an expensive overhead. To address this problem, a new concept, the write-only ORAM is proposed [21], in which only the privacy of the write access pattern is protected.

Unlike the traditional ORAM schemes which perform well based on a binary tree [22,23], when setting asides the privacy of the read access pattern, data structure in the write-only ORAM schemes can be simplified as a flat array, such as HiVE and DetWoORAM [19,24]. HiVE is a write-only ORAM

proposed by Blass et al. to protect the privacy of Hidden Volume Encryption on a disk. In the write operation, a data block is written to the hidden volume uniformly at random. Clearly, the data block transmitted between the client and the server is $O(1)$ in a write operation. However, HiVE stores the position map recursively on the disk. If setting the block size in these recursive levels to $O(logN)$, the overhead of communication in a write operation is $O(log^3N)$ or $O(B)$, where $N$ is the number of data blocks and $B$ is the size of data block. DetWoORAM is a stateless write-only ORAM proposed by Roche et al. In the write operation, the data block is sequentially appended into a holding area on the server. Then, a refreshment is executed to evict the data blocks in the holding area to the main area. In DetWoORAM, a pointer-based technique is used to store the position map and the overhead of communication in a write operation is $O(BlogN)$.

By setting the data block size $B = \Omega(log^2N)$, both schemes have $O(BlogN)$ communication complexity. Moreover, both schemes require $O(logN)$ communication rounds to accomplish a write operation. However, if setting aside the position map, the overheads of communication in both schemes achieve $O(B)$. Therefore, the cost of the position map dominates the efficiency. In our work, we aim to reduce the cost caused by the position map.

### 1.2. The Quest for ORAMs with Optimal Complexity Brought by Position Map

For $N$ blocks, the size of position map is $O(NlogN)$, which is prohibitively large to be stored at the client side. One solution of the previous write-only schemes to address it is the recursive position-based technique, such as HiVE. Concretely, a series of ORAMs such as $ORAM_0$, $ORAM_1$, ... , $ORAM_X$ are constructed where $ORAM_0$ stores the data blocks and the position map of $ORAM_i$ is stored in $ORAM_{i+1}$. The client stores the position map of $ORAM_X$. If the block size in these position map ORAMs is $\chi \cdot logN$ for some constant $\chi \geq 2$, $O(logN)$ recursive levels are constructed until the final ORAM is small enough to fit for the client storage. Although HiVE has a better asymptotic communication complexity than the traditional ORAMs in a non-recursive setting, the costs of position map in the two kinds of ORAM schemes are identical. In the tree-based schemes, both the read operation and write operation must occur for every recursive level, which incurs $O(log^3N)$ communication [22,23]. In Hive, the write operation also occurs for every recursive level. Moreover, when writing at a certain level, all lower recursive levels must be read. Therefore, the overhead of the communication is also $O(log^3N)$ [19].

In addition to the recursively storage, the pointer-based technique is another solution for the storage of position map, in which an oblivious data structure (ODS) in the form of Trie [25] is used to store the position map, such as the DetWoORAM [24]. Every node except the leaves in the Trie stores a pointer to its child node. Instead of a logical address, the physical address of the child is stored in the pointer. Therefore, it is no longer necessary to translate a logical address to a physical address within the Trie itself. Hence, the position map for the Tire itself is eliminated. A path in the Trie is supposed to be traversed when reading the data block or its corresponding physical address. Although the cost of position map is reduced to $O(log^2N)$, it incurs $O(Blog^2N)$ cost to read the data blocks. Moreover, the DetWoORAM requires $O(logN)$ rounds to get the physical address of a block as a node in Trie can only be read from the server after its parent has been read already.

We propose a write-only scheme to improve the efficiency of communication complexity and communication rounds. The construction of position map follows the idea proposed by Gordon et al. [26]. In their scheme, a static position map is used for a two-server-setting ORAM. After initialization, the position map does not change anymore. The privacy of read operation in their scheme is protected by private information retrieval (PIR) [27,28]. In this case, both servers cannot locate the desired data block. The remaining security requirement is identical with that in write-only ORAM. Therefore, the static position map can be employed to our write-only scheme, which brings an overhead of $O(B)$ communication if $B = \Omega(log^2N)$ and $O(1)$ rounds.

### 1.3. On Tightness of Asymptotic Efficiency in Random Write-Only ORAM

In previous write-only ORAM schemes, both the random order and deterministic order are competent to protect the privacy of write access pattern [19,24]. We prefer to the random order, as it generates stash which helps to implement the corresponding write-only ORAM schemes in hidden volume encryption. Additionally, the stash involved could act as an optimization of the communication complexity if the access rates of data blocks are not equal, but this is out of scope of this paper.

HiVE is random write-only ORAM [19], in which the parameters $(r, k)$ manipulate the asymptotic efficiency, where $r$ is the ratios of physical storage to the logical storage. In the evaluation, it only evaluates the stash size on a specific pair of $(r, k)$, without given both the upper bound of stash size and the relationship between $r$ and $k$ to bound the stash size. Based on the mechanism of write access pattern in HiVE, we carry out a tight analysis on the upper bound of stash size. To further reduce the communication complexity, we use the traditional bucket-based data structure to implement our write-only scheme, in which a pair of parameters $(Z, A)$ are defined, where $Z$ is the capacity of blocks in a bucket and $A$ is the rate of a write operation. Parameters $Z$ and $A$ can be traded off to reduce the communication complexity.

### 1.4. Technical Highlights

We construct a two-level ORAM scheme named constant rounds write-only ORAM (CWORAM). The top level stores all data blocks and the second level stores the position map of the data blocks. The position map of the second level is static and does not need to be stored, as the client can compute it at any time.

#### 1.4.1. Construct of the Top Level ORAM

In write-only ORAM schemes, as privacy of a write operation can be achieved by writing a block to a random physical address, a flat array is employed to store blocks in the top level. The dynamic position map of the blocks in the flat array is stored in the second level.

In the flat array, we use the traditional bucket-based structure to store the blocks and set the bucket size as $Z$. In this case, only a bucket is randomly chosen for a write operation. In the analysis, as long as the expected number of blocks and the capacity of blocks included in a write operation are determined, the upper bound of stash size could be evaluated by calculating the deviation.

#### 1.4.2. Construct of Position Map

In this level, it stores the position map of data blocks. Moreover, the position map of the blocks in this level is static. Therefore, a deterministic order has to be used in a write operation of the blocks in this level. We employ a binary tree in this level to support the deterministic order of write operations. There is a requirement of the static position map that the physical address of each block should be randomly distributed. In this case, the stash size can be bounded well. Gordon et al. carry out a pseudo-random permutation (PRP) for each logical block and the result is the physical address of the block. After PRP, the physical addresses of all blocks are randomly distributed.

The main contributions of our work are as follows.

- We use the static position map to reduce the communication complexity of the write-only ORAM. The overhead of communication is reduced from $O(B \log N)$ to $O(B)$, where $B$ is the size of data block and $N$ is the number of data blocks. Moreover, the communication rounds are reduced from $O(\log N)$ to $O(1)$.
- We carry out a theoretical analysis on the stash in our scheme, in which both the upper bound of stash and the parameter-setting condition are taken into account. The analysis expresses that the probability of the stash size exceeding $R$ is $(A/Z)^{-R}$ if $Z(\ln Z/A) - Z + A - \ln 2 > 0$.

The rest of the paper is organized as follows. Section 2 introduces the preliminaries, including the concept of write-only ORAM, the random write-only scheme HiVE, and the technique of static position map proposed by Gordon et al. Section 3 describes our scheme. Section 4 analyzes and evaluates the efficiency of our scheme. Section 5 gives a conclusion.

## 2. Preliminaries

### 2.1. Write-Only ORAM

ORAM was first proposed by Goldreich and Ostrovsky [29] to protect the privacy of access pattern on the RAM. Later, it was used to the client–server scenario, in which the server is treated as honest but curious and the client is trustful. Traditionally, the access pattern protected by ORAM contains the logical address and the type of operation, read, or write. First, the logical address is translated into a group of physical addresses. Second, no matter the logical operation is read or write, it is always translated into a group of physical reads and physical writes.

**Definition 1. (ORAM Security):** *Let physicaladd&op$(\vec{y})$ represents the access pattern containing the physical addresses and operations translated by ORAM from access sequence $\vec{y}$. ORAM is secure as long as, for any two access sequences with the same length, e.g., $\vec{y}_1$ and $\vec{y}_2$, the two access patterns output by ORAM are computationally indistinguishable.*

$$physicaladd\&op(\vec{y}_1) \approx_c physicaladd\&op(\vec{y}_2)$$

*where $\approx_c$ denotes computational indistinguishability (with respect to a computational security parameter $\kappa$).*

The write-only ORAM is a relaxed security notion of ORAM. It sets aside the read access pattern and only protects the write access pattern.

**Definition 2. (Write-Only ORAM Security):** *Let physicaladd$(\vec{y})$ denote the physical addresses translated from access sequence $\vec{y}$ by a write-only ORAM. For two access sequences $\vec{y}_1$ and $\vec{y}_2$ with the same length, it holds*

$$physicaladd(\vec{y}_1) \approx_c physicaladd(\vec{y}_2)$$

### 2.2. HiVE

HiVE is designed to protect the encrypted hidden volume from powerful adversaries which are possible to get multiple-snapshot of a disk and infer the existence of encrypted hidden volume according to the updated locations in the disk.

Data blocks in HiVE are stored in a flat array as the form $(a, d) \in \{0,1\}^{logN} \times \{0,1\}^B$, where $a$ is the logical address of a block and $d$ is the payload. At the client side, there is a stash storing the blocks which have not been written back to the disk yet.

The data structure and the write operation in each recursive level are identical. Specifically, the blocks in each level are stored in a flat array. In a write operation, $k$ blocks are read uniformly at random from the server and the blocks in the stash are inserted into the free spaces of the $k$ blocks if existing any.

In the evaluation, it sets parameters $(r, k) = (2, 3)$ to bound the stash size. The size of block in all position map ORAMs is set to $O(logN)$. Hence, there are $O(logN)$ recursive levels and the communication complexity in position map is $O(log^3N)$. When the data block size is $B = \Omega(log^2N)$, the overhead of communication complexity in HiVE is $O(BlogN)$ and the communication rounds is $O(logN)$.

### 2.3. The Static Position Map

The static position map first proposed by Gordon et al. avoids recursive storage of position map for two-setting setting ORAM. In the initialization, each data block is randomly mapped to a path in a binary tree and the path does not change anymore. In this case, both the read operation and the write operation can be accomplished just with constant communication rounds.

For a logical access, the client first computes the physical path to which the desired block is mapped. Then, the private information retrieval (PIR) implemented by Function Secret Sharing [30] is used to read the desired block from the two servers. On each sever, it takes the entire list of blocks as the inputs of PIR and outputs the secret share of the desired block. Then, the client adds the desired block into the stash. Meanwhile, a path chosen by the reverse lexicographic order [31] are read from one of the servers and the blocks in the stash are written back into it with a greedy algorithm.

There is a requirement for the static position map. That is, each block must be randomly mapped to a physical path, otherwise the stash size cannot be bound. Therefore, the position map is constructed by a pseudo-random permutation $F_K$: $[N] \rightarrow [N]$, with $K$ managed at the client side.

## 3. CWORAM

In this section, we describe the detail of CWORAM, in which only the privacy of write access pattern is protected. For a read operation, the data block and its position map are directly read according to their physical addresses. Notations in CWORAM are listed in Table 1, in which the main ORAM is the top level ORAM and the posMap ORAM is the second level ORAM.

**Table 1.** Notations in CWORAM.

| Notations | Meanings |
| --- | --- |
| $N$ | Number of blocks in the main ORAM |
| $B$ | The size of data block in the main ORAM |
| $Z$ | Maximum number of blocks per bucket in the main ORAM |
| $A$ | The rate of write operation in the main ORAM |
| $S$ | Maximum number of blocks per bucket in the posMap ORAM |
| $A'$ | The rate of write operation in the posMap ORAM |
| $\mathcal{M}_i$ | The *i-th* bucket in the main ORAM |
| $\mathcal{P}_l$ | Path $l$ in the posMap ORAM |
| $\mathcal{P}_{(l,i)}$ | The *i-th* bucket on $\mathcal{P}_l$ in the posMap ORAM |

### 3.1. Data Structure

Figure 1 shows the detail of the data structure in CWORAM. At the top level, the ORAM is filled with data blocks and the data structure is a flat array. We call it the main ORAM, in which each element is a bucket, which possesses a capacity of $Z$ blocks. Data blocks are formed as $(a, d) \in \{0,1\}^{logN} \times \{0,1\}^B$, where $a$ is the logical address and $d$ is the payload.

At the second level, the ORAM is filled with position map of the main ORAM. We call it the posMap ORAM. The data structure in the posMap ORAM is a binary tree, in which each node is a bucket that can hold $S$ blocks. The blocks are formed as $(a, pos(a), off(a)) \in \{0,1\}^{logN} \times \{0,1\}^{logN} \times \{0,1\}^{logZ}$, where $a$ is the logical address of a data block, $pos(a)$ is the physical address indicating where the bucket contains block $a$ is located in the main ORAM, and $off(a)$ is the offset of block $a$ in the bucket. In our scheme, the buckets in both the main ORAM and the posMap ORAM only contain real blocks even if they are not fulfilled, since the dummy block only contributes to the privacy protection of read access pattern [23].

At the client side, it contains two stashes and the key related information. Due to the randomness in the write operations of both the two ORAMs, stashes are required to store the blocks which have not yet been written back to the server. We define *MStash* as the stash in the main ORAM and *PStash* as the stash in the posMap ORAM. In addition, the client stores key related information.

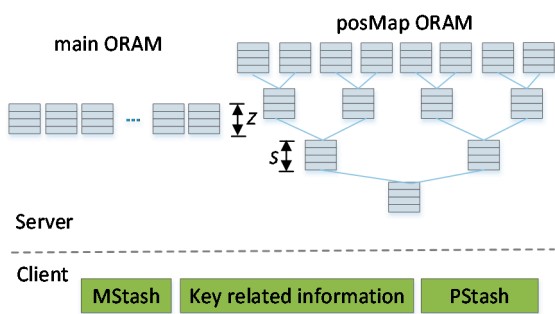

**Figure 1.** Data structure in CWORAM.

## 3.2. Initialization

Data blocks can be stored anywhere in the main ORAM since their addresses are located by the position map. For simplicity, we let one bucket store exactly one block in the main ORAM. Hence, a flat array with $N$ buckets is required. Concretely, the block $a$ is stored in the bucket $\mathcal{M}_a$ and the rest space of bucket $\mathcal{M}_a$ is empty.

In the posMap ORAM, it stores the position map of each block in the main ORAM. The position map of the posMap ORAM is static which is computed by the client directly if needed. Therefore, the recursive storage of position map is unnecessary anymore. For instance, the physical address of data block $a$, i.e., $(a, pos(a), off(a))$, is stored in the posMap ORAM, and the physical address of $(a, pos(a), off(a))$ is computed by the client. Moreover, the physical address of $(a, pos(a), off(a))$ is the index of a leaf node in the posMsp, which indicates that $(a, pos(a), off(a))$ is stored along the path from the root to this leaf node.

To bound the stash size in posMap ORAM, it requires that the physical addresses of all position blocks $(x, pos(x), off(x))$ is randomly distributed, where $x = 0,1, \ldots , N\text{-}1$. Similar to Gordon's scheme, a pseudo-random permutation is used to randomly map each position block to a physical address. Precisely, let $F$ be a pseudo-random permutation and $F_K:[N] \rightarrow [N]$ be a mapping from a position block to its physical address, where $K$ is only known by in the client. Then, the position block $(a, pos(a), off(a))$ is always stored along the path $\mathcal{P}_{F(K,a)}$ in the posMap.

At the beginning, since all blocks in the main ORAM are stored according to their logical addresses explicitly, the posMap ORAM is set to empty.

## 3.3. ORAM Read

As the privacy of the read access pattern does not need to be protect, ORAM Read takes a logical address $a$ as input and responds with block $a$ to the client directly. The detail of ORAM Read is shown in Algorithm 1. First, path $\mathcal{P}_{F(K, a)}$ is firstly read from the posMap ORAM. According to the permutation defined in the initialization, $pos(a)$ and $off(a)$ can be found in path $\mathcal{P}_{F(K, a)}$. Then, block $a$ is read from bucket $\mathcal{M}_{pos(a)}$ with the offset $off(a)$ in the main ORAM. Note that all blocks stored in the server are encrypted and the client needs to decrypt them. The decryption is treated as a default operation and so is not included in the algorithm.

---

**Algorithm 1: ORAM Read($a$)**

---

1: $(pos(a), off(a)) :=$ read $\mathcal{P}_{F(K,a)}$ from posMap ORAM and choose the position of block $a$.
2: $(a, d) :=$ read the block with the offset $off(a)$ in bucket $\mathcal{M}_{pos(a)}$ from the main ORAM.
3: Return block $(a, d)$ to the client

---

### 3.4. ORAM Write

The ORAM Write carries out a write operation and the detail is illustrated in Algorithm 2. The inputs are the block $a$ responded in ORAM Read and its new payload, i.e., $((a, d), d')$. First, the payload of block $a$ is replaced with the new payload $d'$. Then, block $a$ is re-encrypted and put into *MStash*. Afterwards, the sub process `Mwrite` is executed to write the blocks stored in *MStash* to the main ORAM. In the end, since the physical address of each block involved in the `Mwrite` is changed, the sub process `Pwrite` is executed to accomplish the update in the posMap ORAM. Similar to the classical schemes, both the `Mwrite` and `Pwrite` are invoked at fixed rates. We denote the two configurable rates as $A$ and $A'$ for `Mwrite` and `Pwrite`, respectively. Furthermore, a persistent variable *ctr* is set as a counter.

---

**Algorithm 2: ORAM Write($(a, d), d'$) Persistent Variables *ctr*:=0**

---

1: $d:=d'$, $ctr:=ctr+1$
2: $MStash: = MStash \cup (a, d)$
3: **if** $ctr\%A = 0$ **then** `Mwrite()`
4: **if** $ctr\%A' = 0$ **then** `Pwrite()`

---

### 3.4.1. MWrite

After every $A$ `ORAM Reads`, MWrite is executed to write blocks from *MStash* into the main ORAM. A bucket is chosen from the server to find free spaces. Then the blocks in *MStash* are written back into these free spaces. The information hidden from adversaries per `Mwrite` includes (1) which block is being written back (2) the lifetime of the block being written back. To grant the two requirements, both a deterministic order and a random order can be used to choose a bucket. As mentioned above, we employ the random order to set a rate for the MWrite operation.

Suppose bucket $b$ is chosen uniformly at random. There might be two states for its included blocks, fresh or stale. Only the fresh blocks are kept, and the stales are deleted to free up the spaces. The states of these blocks are obtained by querying the posMap ORAM. For block $a$, if its physical address $pos(a)$ stored in posMap ORAM equals the address of bucket $b$, then it is fresh. Otherwise, it is stale.

Afterwards, the client chooses the blocks from *MStash* to insert them into the free spaces of bucket $b$ with either a random order or a sequential order. Since dummy blocks are not included in our scheme, no padding is required even if there still exists free space in bucket $b$. After re-encryption, the bucket $b$ is sent back to the server. Finally, the new physical address of each block written into bucket $b$ are added into *PStash*. The detail of `Mwrite` is described in Algorithm 3. All blocks written back to the server are re-encrypted, which is also treated as a default operation and so is not included in the algorithm.

### 3.4.2. PWrite

After every $A'$ `ORAM Reads`, PWrite is executed to write blocks from *PStash* into the posMap. A path $\mathcal{P}_{\mathcal{W}}$ is chosen to be written according to the reverse lexicographic order of persistent variable *ctr*. First, the path $\mathcal{P}_{\mathcal{W}}$ is read from the server. Then, the stale blocks are cleared out and the fresh blocks are added into *PStash*. In the end, a greedy algorithm is executed to push the blocks in *PStash* down to the path $\mathcal{P}_{\mathcal{W}}$ as far as possible. Specifically, for a block $(a, pos(a), off(a))$, the bucket furthest from the root along path $\mathcal{P}_{\mathcal{W}}$ is found as long as (1) the bucket is on $\mathcal{P}_{F(K, a)}$, and (2) the bucket has free space. Algorithm 4 shows the detail of `Pwrite`.

---

**Algorithm 3: Mwrite (*MStash*)**

---

1: *addr*:=UniformRandom(1,*n*), *b*:= $\mathcal{M}_{addr}$
2: **for** *(a,d)* **in** *b*
3:　*pos*:=read $\mathcal{P}_{F(K,.A)}$ from posMap ORAM and choose the address of block *a*.
4: **if** *pos≠addr* **then** $\mathcal{S}$(BA):=free
5: **while** *b* has free space and *MAtash≠null*
6:　put *blocks* into *b* and delete them from *MStash*
7: **for** *(a,d)* **in** *blocks*
8:　PStash:= PStash∪(a, addr)

---

---

**Algorithm 4: Pwrite (*PStash*)**

---

1: $\mathcal{P}_{W}$:=the path determined by the reverse lexicographic order of *ctr/A'*.
2: Clear out the stale block in $\mathcal{P}_{W}$ and put the rest fresh blocks into *PStash*
3: **for** *l* from *logN*-1 **to** 0
4:　**if** $\mathcal{P}_{W,L,}$ has free space and *PStash ≠null*
5:　**then** choose blocks which can reside in bucket $\mathcal{P}_{W,l,}$ and delete them from *PStash*

---

*3.5. Security Analysis*

### 3.5.1. Encryption Mode

At any time, data blocks and their position map stored on the server are encrypted. To ensure the semantic security for all data blocks and their position map, an encryption scheme against chosen-plaintext attacks (CPA) is used in our scheme. CPA security is often called IND-CPA security which products indistinguishable ciphertexts from random strings, such as the Advanced Encryption Standard (AES) in Cipher Block Chaining (CBC) or counter mode. In this case, the server cannot learn any information from the ciphertexts stored on it.

### 3.5.2. Security of CWORAM

In the initialization, all data blocks are stored in the main ORAM after encryption. In addition, the key used for static mapping is managed by the client. Hence, no information is leaked to the server.

In the sub process `Mwrite`, the sequence of physical addresses written to the main ORAM are random, which does not depend on the access sequence. Furthermore, the sequence of physical paths written to the posMap are deterministic, which also does not depend on the access sequence. In addition, the IND-CPA encryption mode is used to encrypt the data. Hence, the server is unable to get any information from the ciphertexts. Therefore, let $\vec{y}_1$ and $\vec{y}_2$ be two access sequences with the same length, it holds

$$physicaladd(\vec{y}_1) \approx_c physicaladd(\vec{y}_2)$$

## 4. Evaluation

In our evaluation, we set the block size $B = \Omega(log^2N)$ in the main ORAM and the block size $B' = \Omega(logN)$ in the posMap. In the following, we evaluate the storage in both the client and server, as well as the communication complexity of the write operation.

*4.1. Server Storage*

All the blocks are encrypted on the server with a computational security parameter $\kappa$. In the main ORAM, there are *N*-buckets which have the capacity of *Z* blocks. Encryption adds less than $\kappa$ additional bits to each block. When the block size is $B = \Omega(log^2N)$, the length of encrypted block is $D_M=O(log^2N)$, which results $O(N \cdot log^2N)$ server storage. In the posMap ORAM, there are *N*-buckets that have the capacity of *S* blocks. Similar to the main ORAM, the encryption adds less than $\kappa$ additional

bits to each block. When the block size is $B' = \Omega(logN)$, the length of encrypted block is $D_P = O(logN)$, which results $O(N \cdot logN)$ server storage. Therefore, the server storage could be concluded as $O(N \cdot B)$.

*4.2. Communication Complexity*

A logical read from the client is translated into an ORAM Read, which responds with a block to the client. The physical reads in ORAM Read include (1) a block read from the main ORAM, and (2) a path read from the posMap ORAM. Hence, the communication complexity for a logical read is $D_m + S \cdot \log N \cdot D_p$ .

A logical write from the client is translated into an ORAM Read and an ORAM Write. The ORAM Write writes the blocks from the client to the server. First, a physical read is executed in `Mwrite` to read a random bucket from the main ORAM. Then, to check the states of the blocks in the random buckets, at most $Z$ paths are physically read from the posMap ORAM in `Mwrite`. Afterwards, a physical write is executed to write the random bucket back. In the end, both a physical read and physical write of a deterministic path are executed in `Pwrite` to update the position map of the blocks in the main ORAM. Comparing a logical read, the extra communication complexity for a logical write is $2Z \cdot D_M / A + Z \cdot S \cdot \log N \cdot D_p / A + 2S \cdot \log N \cdot D_p / A'$.

As $D_M=O(B)$ and $D_p=O(logN)$, the total communication complexity is $O(B)$ or $O(log^2N)$. If $B=\Omega(log^2N)$, the total communication complexity per logical write or logical read is $O(B)$. When $N$ and $B$ are fixed, we can trade off $Z$ and $A$ to reduce the communication complexity in the main ORAM. Likewise, $S$ and $A$ can be traded off to reduce the communication complexity in the posMap ORAM.

Furthermore, for a logical read, the physical reads in ORAM Read can only be executed sequentially, which results two communication rounds. For a logical write, ORAM Read and ORAM Write can be executed in parallel. In ORAM Write, the first communication round is the physical read to a random bucket and the second communication round is the physical read to a set of paths in posMap ORAM to check the states of blocks in the random bucket. For the remaining operations, all of them can piggyback on the two physical reads. Specifically, the physical write to the random bucket as well as the physical read and physical write to update the position map could be executed together with the following physical reads to a random bucket or a set of paths. As a result, two communication rounds are required for a logical write.

*4.3. Stash Size*

Due to the randomness of `Mwrite` and `Pwrite`, both the main ORAM and the posMap ORAM require a stash to store the blocks which have not been written back yet. Since the detail analysis on tree-based ORAM has been discussed by Gordon et al. [29], we only make a brief discussion on it, but we will carry out a detail analysis of the stash size in the main ORAM.

4.3.1. PosMap ORAM

First, the scheme $ORAM_{\infty}^{P,L}$ with infinite bucket size is introduced. The original scheme with bucket size $S$ is denoted as $ORAM_S^{P,L}$. After running the same access sequence with the same randomness in the two schemes, $ORAM_{\infty}^{P,L}$ after a post-process algorithm has exactly the same state with $ORAM_S^{P,L}$. Specifically, the two schemes have the same distribution of blocks over all buckets and the stash. For any rooted subtree $T$ in $ORAM_{\infty}^{P,L}$, if the number of blocks in $T$ (denoted as $X(T)$) is larger than the capacity of the same subtree in $ORAM_S^{P,L}$, the extra blocks in $T$ are the ones stored in *PStash*. Using a union bound and Catalan sequence bound, the probability that the size of *PStash* is more than $R$ is as follows, where $n(T)$ is the number of nodes in subtree $T$.

$$\Pr[PStash \geq R] < \sum_{n \geq 1} 4^n \cdot \max_{T:n(T)=n} \Pr[X(T) \geq n(T) \cdot S + R] \tag{1}$$

The bound of expected block number in any subtree $T$ ($E[X(T)]$) is calculated firstly. Following the analysis of Gordon et al. ([26], Section 3.2), we have $E[b_\infty] \leq 1$ for the buckets in leaf nodes and $E[b_\infty] \leq A'/2$ for other buckets.

In the end, based on the capacity of each rooted subtree $n(T) \cdot S$ and the expected number of blocks $E[X(T)]$ in $ORAM_\infty^{P,L}$, a Chernoff-like bound is used to analysis the size of *PStash*. If $N \geq 4$ and $A'=1$, it holds $E[X(T)] \leq 0.8 \cdot n(T)$, which brings a conclusion that as long as $S \geq 3$, the *PStash* overflow probability decreases exponentially in its size $R$. Table 2 shows the extrapolated the size of *PStash* for different $(S, A')$ with overflow probability of $2^{-\lambda}$, where $\lambda$ is a statistical security parameter.

**Table 2.** The bound of *PStash* with the overflow probability $2^{-\lambda}$.

|  | (S,A') | | | |
|---|---|---|---|---|
| $\lambda$ | (3,1) | (4,3) | (5,4) | (6,5) |
| 40 | 15 | 32 | 33 | 34 |
| 80 | 37 | 63 | 64 | 65 |
| 128 | 67 | 91 | 93 | 94 |

### 4.3.2. Main ORAM

The analysis of stash size in the main ORAM could follow the idea proposed by Path ORAM [13]. Similar to the posMap ORAM, $ORAM_\infty^{M,L}$ is defined in which the bucket size is infinite and $ORAM_Z^{M,L}$ is defined in which the bucket size is $Z$. First, the states of $ORAM_Z^{M,L}$ and $ORAM_\infty^{M,L}$ after a post-process algorithm are discussed. Then, the overflow probability of $ORAM_Z^{M,L}$ is analyzed. Afterwards, the influence of access sequence on the stash size is analyzed. Finally, the bound of stash size is calculated.

Let $b_\infty$ be a bucket in $ORAM_\infty^{M,L}$ and $b_Z$ be a bucket in $ORAM_Z^{M,L}$. Suppose $ORAM_\infty^{M,L}$ and $ORAM_Z^{M,L}$ are running the same access sequence with the same randomness. A post-process of $ORAM_\infty^{M,L}$ is carried out as follows. The buckets involved in ORAM Write are visited one-by-one. At time stamp $i$, if $b_{\infty,i}$ has more blocks than $b_{Z,i}$, the extra blocks are stored in the client. If $b_{\infty,i}$ has less blocks than $b_{Z,i}$, the blocks which do not reside in $b_{\infty,i}$ must be stored in the client. Then, these less blocks are stored in $b_{\infty,i}$. Therefore, the state of $ORAM_\infty^{M,L}$ after post-process is identical to $ORAM_Z^{M,L}$.

Let $T$ represents a sequence of buckets randomly chosen by ORAM Writes in $ORAM_\infty^{M,L}$, $X(T)$ denote the number of fresh blocks in $T$, and $n(T)$ denote the number of buckets in $T$.

**Lemma 1.** *The MStash size is more than R if and only if there exists a sequence of buckets T such that X(T)> Z·n(T)+R.*

If there exists a sequence of buckets $T$ such that $X(T) > Z \cdot n(T) + R$, then the client storage generated by post-process of $ORAM_\infty^{M,L}$ is larger than $R$. As the *MStash* has the same state with the client storage generated by post-process of $ORAM_\infty^{M,L}$, the *MStash* size in $ORAM_Z^{M,L}$ is also larger than $R$. Otherwise, the *MStash* size is no more than $R$.

For $n$ buckets, it contains the most numbers of $n(n+1)/2$ sequences mentioned in Lemma 1. By using a union bound, we have

$$\begin{aligned} \Pr[MStash \geq R] = & \ \Pr[\exists T \in ORAM_\infty^L, X(T) \geq Z \cdot n(T) + R] \\ < & \sum_{n \geq 1} n(n+1)/2 \max_{T:n(T)=n} \Pr[X(T) \geq Z \cdot n(T) + R] \end{aligned} \tag{2}$$

**Lemma 2.** *For all access sequences constructed by N blocks, the probability $Pr[X(T) \geq Z \cdot n(T) + R]$ is maximized under the access sequence in which the logical address of each block only appears exactly once.*

Suppose there is a logical address in access sequence a that has been written twice. Then, there exists two indices i and j, i < j, such that $a_i = a_j$. Let T be the sequence of buckets such that X(T) is maximized under sequence a. Without the j-th ORAM Write, X(T) remains the maximum. On the contrary, if there is no duplicated logical address in an access sequence, then the j-th logical address is distinct with other logical addresses. In the j-th ORAM Write, if the j-th logical block is stored in T, the maximum number of blocks in T becomes X'(T) = X(T)+1, as the bucket size is infinite. Therefore, the access sequence in which each logical address only appears once maximizes the stash size.

If the maximum of X(T) generated by the access sequence mentioned in Lemma 2 has an upper bound, *MStash* is bounded. Let $X_i(T) \in \{0,1\}$ indicates whether the i-th accessed block is in T and $p_i = \Pr[X_i(T) = 1]$. As the uniform randomness, the probability that each bucket is chosen in ORAM Write is 1/N. For a N-length access sequence, we have $E[X(T)] \leq n(T) \cdot A$. Since the logical address of each block is distinct, it means that $X_i(T)$, i = 0,..., N-1 are all statistically independent. According to Ring ORAM ([2], Section 4.3), the generate function of $E[e^{tX(T)}]$ is bounded as $e^{nA(e^t-1)}$, where n = n(T) and t > 0. By the Markov Inequality, the Chernoff-bound on X(T) is

$$\begin{aligned} \Pr[X(T) \geq nZ + R] = \quad &\Pr[e^{X(T)} \geq e^{nZ+R}] \\ &\leq E[e^{tX(T)}] \cdot e^{-t(nZ+R)} \leq e^{(e^t-1)An} \cdot e^{-t(nZ+R)} \\ &= e^{-tR} \cdot e^{-n(Zt-A(e^t-1))} \end{aligned} \tag{3}$$

From Equations (2) and (3), the probability that the size of MStash exceeds R is

$$\Pr[MStash \geq R] \leq \sum_{n=1} n(n+1)/2 \cdot e^{-tR} \cdot e^{-n(Zt-A(e^t-1))} < \sum_{n \geq 1} e^{-tR} \cdot e^{-n(Zt-A(e^t-1)-\ln 2)} \tag{4}$$

The above inequality holds since $n(n+1)/2 \leq 2^n$ when n≥1. Let t = lnZ/A, as long as $q = Z \ln Z/A - Z + A - \ln 2 > 0$, the overflow probability of *MStash* is $(A/Z)^R \cdot e^{-q} \cdot (1 - e^{-q})$, which decreases exponentially in its size R.

The analysis shows that the size of *MStash* only relies on parameters (Z, A) and the number of blocks N has little influence on it. We simulate our scheme for a single long run. The access sequence is designed in a round-robin pattern from Lemma 2, i.e., {1,2,...,N,1,2,...,N,1,2,...}. In the initialization, we store number of N distinct data blocks to the main ORAM according to their logical addresses, e.g., block a is stored in $\mathcal{M}_a$.

We ran $2^{25}$ logical writes and the first N logical writes were used to warm up the main ORAM to a steady state. The parameters (Z,A) are set to (3, 1). Figure 2a shows the size of MStash under different N. Four curves with different values of statistical security parameter λ are analyzed. Each curve represents the size of MStash by varying the number of blocks (N), which concludes that N has little impact on the size of MStash.

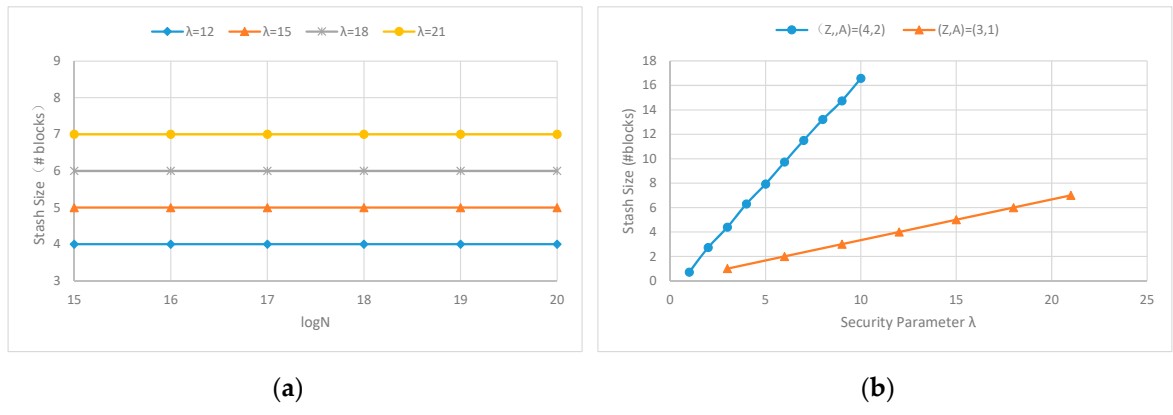

(a)  (b)

**Figure 2.** (**a**) Maximum size of *MStash* with the failure probability $2^{-\lambda}$ under different *N*; (**b**) Maximum size of *MStash* with the failure probability $2^{-\lambda}$ under different (Z, A), N = $2^{15}$.

By setting $N = 2^{15}$, Figure 2b plots the size of *MStash* is no more than $R$ with a failure probability $2^{-\lambda}$. The two curves correspond to $(Z, A) = (3, 1)$ and $(Z, A) = (4, 2)$. Both curves are linear, which shows that the size of *MStash* exceeds R with probability of $2^{-cR}$, where $c$ is a constant depending on $(Z, A)$.

We have analyzed and evaluated the client storage required by both the main ORAM and posMap ORAM. The results show that the client storage required in posMap ORAM is $\omega(log^2 N)$ and the client storage required in main ORAM is $O(B)$. Since $B = \Omega(log^2 N)$, the total client storage can be concluded as $\omega(B)$.

Table 3 compares the efficiency of CWORAM with those of the previous write-only ORAM schemes, as well as that of Gordon's. For a logical write operation, the previous works have the minimum communication complexity $O(B log N)$. In CWORAM, since neither the recursive storage nor the ODS storage of position map is required, it achieves $O(B)$ communication complexity if the size of blocks in the posMap is set to $O(log N)$. For a logical read operation, the overhead of communication in CWORAM is the same with HiVE, i.e., $O(B)$, which improves a factor of $log N$ comparing to that in DetWoORAM. Further, CWORAM brings a constant communication rounds which is the minimum rounds achieved by the previous works. Moreover, CWORAM also performs well in terms of the storage of the stash and that of the server.

**Table 3.** The efficiency of ORAMs in a uniform block setting, i.e., $B = \Omega(log^2 N)$.

| Parameters / Schemes | Communication | | Rounds | | Stash | Server Storage | Security |
|---|---|---|---|---|---|---|---|
| | Logical Read | Logical Write | Logical Read | Logical Write | | | |
| HiVE | $O(B)$ | $O(B log N)$ | $O(log N)$ | $O(log^2 N)$ | $\omega(B)$ | $O(BN)$ | W |
| DetWoORAM | $O(B log N)$ | $O(B log N)$ | $O(log N)$ | $O(log N)$ | $O(B)$ | $O(BN)$ | W |
| Gordon's | $O(B log N)$ | $O(B log N)$ | $O(1)$ | $O(1)$ | $\omega(B log N)$ | $O(BN)$ | R&W |
| CWORAM | $O(B)$ | $O(B)$ | $O(1)$ | $O(1)$ | $\omega(B)$ | $O(BN)$ | W |

## 5. Conclusions

By leveraging the property that the position map can be static, we propose a write-only ORAM called CWORAM with $O(B)$ communication complexity. Moreover, for each logical write, the communication is accomplished with a constant number of rounds. Additionally, the storages required in both the server and the client are bounded well, with $O(NB)$ and $\omega(B)$, respectively. As our scheme contains a stash, it increases the management complexity in the cloud storage synchronization. However, as aforementioned, the stash could be used to further reduce the communication complexity if allowing it to store the high-frequency data blocks. Therefore, in our future work, we will study the trade-off between the stash size and the communication complexity under different update rates of data blocks.

**Author Contributions:** Conceptualization, B.Z. and Z.C.; methodology, H.L.; software, Z.C.; validation, B.Z., Z.C., and H.L.; formal analysis, B.Z.; investigation, M.J.; resources, B.Z. and X.J.; data curation, B.Z.; writing—original draft preparation, Z.C.; writing—review and editing, H.L.; visualization, B.Z.; supervision, B.Z.; project administration, B.Z. and H.L.; and funding acquisition, B.Z. All authors have read and agreed to the published version of the manuscript.

**Funding:** This research was funded by the Wuhan FRONTIER Program of Application Foundation under Grant 2018010401011295 and the Wuhan Applied Basic Research Program of Foundation under Grant 2017010201010117.

**Acknowledgments:** Thanks for all the advisors and colleagues who support our work.

**Conflicts of Interest:** The authors declare no conflict of interest.

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
