# Peer review of "A Constant Round Write-Only ORAM"

_applsci, doi:10.3390/app10155366_

Round 1

Reviewer 1 Report

This submission proposes an oblivious RMA (ORAM) which protects an information in writing a data into the ORAM.

The main concern is a lack of discussion on the security. The authors should give a security goal (for example, the authors use terms, oblivious and trivial, without their meanings) and an adversarial model, and explain why the ORAM is secure by giving a proof.

The second concern is a lack of discussion on the technical improvement of their scheme from the previous researches. The proposed ORAM seems just a combination of two kinds of ORAMs, a flat ORAM and a binary tree-based ORAM. There are lots of researches on the ORAM as we can find in the Internet. I am not sure that there is no research which discusses such a combination.

Minor comments:

Line 115. “private retrieval information” -> “private information retrieval”

Table 2. “N | Number of blocks …” seems “N | Number of buckets …”

Line 214. The authors should give a definition of “B” (by adding it in Table 2 etc.).

Line 236. The authors should explain what the “path” is.

The section titles of 3.4 and 3.5 are identical. The authors should use different titles which summarizes each contents.

Author Response

Thanks for your review, your comments help a lot to improve the quality of our manuscript. We have addressed all your comments in the revised version of this paper.

Reviewer 2 Report

The paper is well structured.

The  topic is not of a hot interest however the paper is of a correct level.

Author Response

Thank you to review our manuscript and approve our work.

Reviewer 3 Report

The paper presents the use of a write-only ORAM to efficiently protect the privacy of applications such as cloud storage synchronization and encrypted hidden volumes.

As a general comment, I think the paper makes an interesting contribution to the literature. At the same time, I think the paper needs minor improvements before to be published in this journal.

Broad comments

  • The introduction should be enriched. For example, in line 96 some references should be added to support the sentence.
  • The authors should describe better if the study is a theorical or empirical investigation.
  • I would suggest the authors to add a subsection in the section 4 “Evaluation” where a comparison among CWORAM and previous ORAM is showed quantitatively.
  • I would suggest the authors to enrich the conclusion section with limitation and next steps
  • The references should be ordered according to the text citations

Author Response

(The authors gave the same response as above.)

Round 2

Reviewer 1 Report

Minor comments 

Line 131, period is required (ln 2 > 0.).

Line 146, it seems “ORAM is secure as...”.

Line 174, period is required (et al.).

Table 1, the horizontal line between second and third rows should be a single line (¥hline, not ¥hline¥hline).

Author Response

Sorry for our negligence.  We have modified them in the revised version.

Thanks for your helps.